# Primary Renal Angiosarcoma: Rare Tumour with Lethal Outcomes

**DOI:** 10.3390/medicina60060885

**Published:** 2024-05-28

**Authors:** Zilvinas Venclovas, Aurelija Alksnyte, Urte Rimsaite, Tomas Navickis, Mindaugas Jievaltas, Daimantas Milonas

**Affiliations:** 1Department of Urology, Medical Academy, Lithuanian University of Health Sciences, 44307 Kaunas, Lithuania; 2Faculty of Medicine, Medical Academy, Lithuanian University of Health Sciences, 44307 Kaunas, Lithuania; 3Department of Pathology, Medical Academy, Lithuanian University of Health Sciences, 44307 Kaunas, Lithuania

**Keywords:** renal haemangioma, primary renal angiosarcoma, nephrectomy, metastasis, diagnosis

## Abstract

*Introduction*: Renal haemangioma is a benign tumour, and due to its characteristics, it must be distinguished from malignant diseases. We present a clinical case of primary renal angiosarcoma initially mistaken for haemangioma due to their similarity. *Case report*: A 58-year-old man was admitted to the hospital with suspicion of pulmonary embolism. The patient complained of pain on the left side. An ultrasound and CT scan of the abdomen showed a tumour mass ~20.5 × 17.2 × 15.4 cm in size in the projection of the left kidney. On CT images, there were data for clear cell renal clear cell carcinoma (ccRCC). A left nephrectomy was performed. However, histological examination revealed renal haemangioma. Three months later, the patient presented to the hospital with abdominal and lumbar pain. A CT scan showed multiple small hypoechoic foci up to 2 cm in size in the liver, lungs, and intra-abdominally, with the most data for carcinosis. Histological re-verification of the left kidney showed a renal vascular tumour with pronounced signs of infarction and necrosis with the majority of the evidence supporting angiosarcoma. Despite treatment, the patient’s outcome was fatal. *Conclusions*: Based on the clinical presentation, radiological images and histological examination data, the tumour was initially misdiagnosed as kidney haemangioma. Due to the rarity of this tumour, there are no established treatment protocols or clinical guidelines for managing primary kidney angiosarcoma.

## 1. Introduction

Renal haemangioma is a rare benign tumour with various clinical and radiological manifestations. Therefore, it is difficult to differentiate this tumour from renal cancer before surgery [1]. Renal haemangioma is histologically characterised by a unique sinusoidal architecture resembling splenic parenchyma, which may raise concerns about angiosarcoma [2]. This makes it difficult, after extensive medical investigations, to differentiate renal haemangioma from malignant diseases, which is why radical surgery is the most common treatment. In addition, haemangiomas are characterised by bleeding, which can range from persistent microhaematuria to continuous bleeding, leading to haemodynamic disturbances. Timely surgical treatment helps to prevent possible spontaneous rupture of the kidney [3,4].

Angiosarcoma is a rare malignant neoplasm that accounts for less than 2% of all soft tissue sarcomas. One-third of angiosarcomas occur in the skin; another third in soft tissues, and the remaining third in bones, breasts, and liver [5]. Meanwhile, primary renal angiosarcoma is extremely rare, accounting for only 1% of all angiosarcomas [6]. Up to 2023, there were only 113 reported cases of primary renal angiosarcoma [7]. The latter renal tumour is rare but highly aggressive, with an unfavourable prognosis [5]. The prognosis depends on the size and stage of the tumour. The 5-year survival rate is 32% in patients with a tumour size of less than 5 cm. In contrast, patients with renal angiosarcoma larger than 5 cm have a survival rate of only 13%. Metastatic disease is an indicator of poor prognosis [8].

We report a clinical case of primary renal angiosarcoma, which was very similar to renal haemangioma and was therefore misdiagnosed.

## 2. Case Description

A 58-year-old man with a hypersthenic body build (height 180 cm, weight 140 kg, BMI 43.21 kg/m^2^) presented to the Emergency Department (ED) with shortness of breath that had been present for 24 h and had been increasing. On physical examination, decreased oxygen saturation (SpO2) of 90% was observed. Elevated D-dimer levels (>50.00 mg/L) were observed in blood tests. Suspecting pulmonary embolism (PE), the patient urgently underwent a chest computed tomography (CT) scan with intravenous contrast agent administration. The CT scan showed mild congestive changes bilaterally in the lungs, which meant that data for PE were insufficient. The patient was hospitalised for observation and further investigation.

After the CT scan with the contrast agent, the patient developed acute renal failure (ARF) (creatinine 112 → 154 µmol/L and eGFR 63 → 42 mL/min) with no effect of infusion therapy. After the investigations, the patient started complaining of pain in the left side at rest, sometimes radiating to the axilla. An ultrasound (US) of the abdominal organs was performed. On examination, a non-homogeneous mass measuring 132 × 134 mm was observed in the left kidney. The mass could not be palpated due to the patient’s high BMI.

The CT scan of the abdomen and pelvis showed a streak of free fluid in the left pleural cavity and a ~20.5 × 17.2 × 15.4 cm tumour mass of heterogeneous structure in the projection of the left kidney. CT images showed the most data for clear cell renal clear cell carcinoma (ccRCC) (Figure 1). The CT scan showed no evidence of remote metastases. Blood tests showed elevated levels of D-dimers (20 mg/L), creatinine (110 µmol/L), CRP (46.1 mg/L), anaemia (HGB 10^9^ g/L), moderate leucocytosis with neutrophilia, and thrombocytopenia (PLT 122 × 10^9^ g/L). The anaemia was increasing (HGB 109 → 97 → 71 g/L). Gastrointestinal bleeding was ruled out on investigation. The CT scan of the abdomen and pelvis was performed because of abdominal pain and progressive anaemia, which showed a massive formation in the left kidney with signs of active bleeding and, also, signs of renal ischaemia. It was decided to perform angiography, which encompassed a renal embolisation, followed by three units of erythrocyte mass transfusion (EMT). With persistent anaemia (HGB—80 g/L), it was decided to administer two more units of EMT and six units of cryoprecipitate.

After the stabilisation of the patient’s condition, open left nephrectomy was performed. Histological examination revealed renal haemangioma with marked signs of necrosis and renal infarction. Sinus histiocytosis was observed in four lymph nodes (Figure 2). The post-operative course was satisfactory, and the patient was discharged as an outpatient on the 7th day after surgery.

Three months after hospitalisation, the patient presented to the ED because of abdominal and lumbar pain. During this period, the patient had lost 30 kg of weight (BMI 43.2 → 33.9 kg/m^2^). The patient had no trauma and did not use any pro-haemorrhagic drugs. The abdominal ultrasound showed multiple, in some places, confluent hyperechogenic foci in the abdomen. A CT scan of the abdomen and pelvis was performed. Ascites and multiple small hypoechoic foci up to 2 cm in size were observed in the liver and lungs, and multiple foci of different sizes were observed intra-abdominally (along the post-operative bed of the left kidney, spleen, stomach, and pancreas), with the most data for carcinosis. Assessing these radiological findings, such spreading of a haemangioma is highly unlikely; therefore, histological verification is required. Histological re-verification of the left kidney was performed by the same uropathologist team. They found a renal vascular tumour with pronounced signs of infarction and necrosis; after further discussion of the patient and evaluation of the negative dynamics of the disease, mitotic activity with focal cellularity and cellular polymorphism, and necrosis of the tissues, the majority of the data were in favour of angiosarcoma.

Due to the abdominal pain and ascites, abdominal paracentesis was performed. Three L of ascites with haemorrhagic fluid was drained within a day. The patient’s renal function deteriorated over time (creatinine 182 → 369 → 592 µmol/L), and anaemia worsened (HGB 109 → 94 → 67 g/L). The patient was treated conservatively. Due to persistent abdominal pain and pronounced anaemia after ascites puncture, suspecting possible bleeding, a CT scan was performed: there was no evidence of active bleeding, but a large post-puncture haematoma was observed. As the patient’s condition was deteriorating and anaemia was worsening, a multi-disciplinary medical consilium of oncologists, surgeons, and radiologists was summoned. Regarding the suspected bleeding, during the discussion, it was decided to perform a diagnostic laparoscopy to find the cause of the bleeding. In the pre-surgical coagulogram, a prolonged PT time of 30.5 s was observed (normal range: 19.8–25.8 s); INR and PTT were normal. The laparoscopy was converted to a laparotomy due to multiple intestinal adhesions. The revision revealed abundant haemorrhagic ascites and carcinomatous foci in the greater omentum, liver, and mesentery of the small intestines, with diffuse bleeding from these carcinomatous foci. No additional surgical procedures were possible. The patient’s condition worsened in dynamics with conservative treatment. The anaemia (HGB 60 g/L) and renal failure (creatinine 861 µmol/L) worsened, and a prolonged PT (29.3 s) was also observed. Despite the treatment, four days after the diagnostic surgical intervention, the patient’s lethal outcome could not be avoided.

## 3. Discussion

Renal haemangioma is a rare tumour of benign origin, usually without significant symptomatology. Preoperative diagnosis of this tumour is difficult and sometimes impossible. The size of the tumour varies from a few to several centimetres. Although haemangiomas can occur anywhere in the body, the kidneys are the most commonly affected [9]. Haemangioma is one of the causes of painless haematuria in people aged 16–24 years [4]. When the tumour is large, the patient may complain of pain in the side of the body and a palpable mass in the abdominal cavity. Patients with sudden onset flank pain, lumbar mass, and signs of internal bleeding (Lenk’s triad) should raise suspicion of Wunderlinch syndrome [10]. Wunderlinch syndrome (WS) refers to spontaneous renal haemorrhage in the subcapsular and perirenal spaces occurring in the absence of known trauma. WS is commonly caused by benign and malignant renal neoplasms that account for up to 60% of all cases. Diagnosis of WS relies on symptoms and a CT scan or MRI. The recommended treatment is surgery or embolisation to control the bleeding [11,12].

Due to its characteristics, haemangioma should be carefully differentiated from malignant tumours [9]. Histologically, irregularly shaped anastomosing sinusoidal spaces lined with minimally atypical endothelial cells are observed, but it is difficult to distinguish haemangioma from angiosarcoma [13]. There are cases in the literature where haemangiomas have been misdiagnosed as RCC or urothelial cell carcinoma [14,15]. Thus, due to the diversity of tumour presentation, even histological examination may be a cause for concern due to its similarity to malignancy [2]. Renal haemangioma findings are usually non-specific and, therefore, mimic other tumours. Microscopically, the typical fusion of vascular channels with endothelial cells forms a network that is diagnostically useful. Hyaline globules in endothelial cells are known to be found in renal haemangiomas. The latter findings may also be detected in other skin and soft tissue tumours such as Kaposi’s sarcoma and angiosarcoma. The difference is that renal haemangioma lesions are usually small in size, dominated by an anastomosing architecture with minimally atypical endothelial cells without evidence of malignancy. In contrast to renal haemangioma, angiosarcoma often presents as a large, necrotic renal lesion with parenchymal invasion. Microscopically, marked cellular infiltration, mitotic activity, necrosis, multilayering, and marked cellular atypia are observed [16].

Angiosarcoma is a rare tumour, occurring in 2–3% of all soft tissue sarcomas. The tumour can occur anywhere in the body, but skin, soft tissue, liver, and bones are the most commonly affected. The kidneys are most often affected by metastases due to systemic spread of the disease. Primary angiosarcoma in the kidney is very rare and associated with a poor prognosis. This tumour is most common in men aged 50–60 years [8]. Early diagnosis is difficult, because there are no symptoms in the early stages of tumour growth, and the disease progresses rapidly [6]. Clinical manifestations include pain in the side of the body, haematuria, anaemia, and a palpable mass [17]. Metastases are frequent and develop within several months after surgical treatment. The most common sites of remote metastases are the lungs, liver, and bones. Clinical and radiological findings are only suggestive of the presence of a tumour, and the exact diagnosis depends on the histopathological examination that is performed after nephrectomy. A positive reaction of endothelial markers with immunohistochemical staining CD-31, CD-34, and factor VIII must be observed. The rarity of the tumour makes it difficult to develop a standardised treatment protocol [6].

As it is a disease that is usually diagnosed after metastases have already been detected, its local treatment is complex, and there is no universally accepted standardised treatment. The prognosis of the patient depends on the size of the tumour and the degree of spread of the disease [7]. One thing is certain: radical nephrectomy should be performed if possible. Some authors recommend adjuvant radiotherapy or chemotherapy, although its benefit is not proven [6,17]. Currently, there are no clear treatment guidelines to guide the choice of chemotherapy regimen for adjuvant treatment of renal angiosarcoma. In a retrospective study, Constantinidou et al. [18] determined that the most common choice of chemotherapy for adjuvant treatment was paclitaxel monotherapy or a combination of gemcitabine and docetaxel. Meanwhile, the standard treatment for metastatic renal angiosarcoma is chemotherapy with cisplatin and ifosfamide, with doxorubicin with ifosfamide as an alternative. However, taxanes (paclitaxel or docetaxel) are thought to be more effective for systemically advanced disease [18,19]. However, this disease has a high mortality rate, with most patients not surviving more than 12 months, and there is a lack of standardised management protocols [7,17,18,19].

## 4. Conclusions

Primary renal angiosarcoma is extremely rare. This clinical case highlights the importance of careful differentiation of renal angiosarcoma from other tumours that may mimic this disease. In this clinical case, based on the clinical presentation, radiological findings, and histological examination, the tumour was initially misdiagnosed as a renal haemangioma. It is important to be aware of the variety of radiological appearances and histological expression of this rare malignant process. Unfortunately, due to the rarity of this tumour, its treatment remains unclear, and clinical guidelines for the treatment of primary angiosarcoma have not been established.

## Figures and Tables

**Figure 1 medicina-60-00885-f001:**
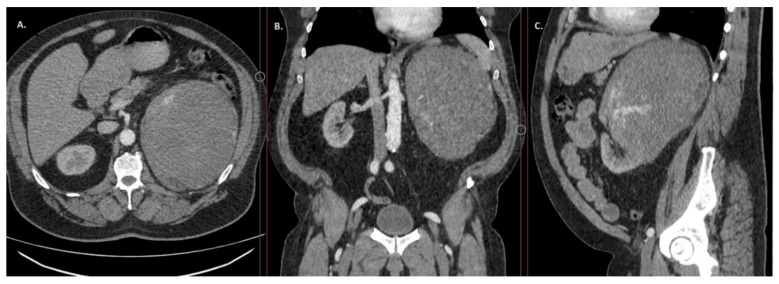
(**A**) Axial, (**B**) coronal, and (**C**) sagittal CT scan shows massive formation in the left kidney.

**Figure 2 medicina-60-00885-f002:**
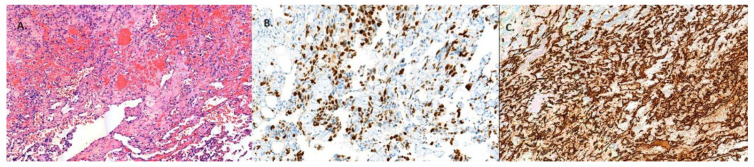
(**A**) Histological sections reveal proliferation of the vessels and anastomosing channels with vasoformation and dissecting growth pattern (H&E stain). (**B**) Ki-67 and (**C**) CD34.

## Data Availability

The raw data supporting the conclusions of this article will be made available by the authors, without undue reservation.

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
