# Peer review of "Primary Renal Angiosarcoma: Rare Tumour with Lethal Outcomes"

_medicina, 2024, doi:10.3390/medicina60060885_

Round 1

Reviewer 1 Report

Comments and Suggestions for Authors

An interesting presentation of the case.

I would advise changing the title: Primary Renal Angiosarcoma: Rare Cancer with Lethal Outcomes to Primary Renal Angiosarcoma: Rare Tumour with Lethal Outcomes.

At the time of the first CT scan, the creatinine concentration?

Line 62: GFR 42 ml/min, is that eGFR? creatinine concentration?

Line 100 after further discussion.....explain with whom? a pathologist?

Line 103 aspiration of 3 liters of ascites, risk of AKI? Explain why 3 liters

Author Response

Dear reviewer,

Thank you for your time and observations. We have made some changes in the main text according to your recommendations. 

We have changed the title.

We have added the level of creatine concentration before the first CT and how it increased eventually. 

Because metastasis was found, the same dedicated uropathologist evaluated the findings again.

Due to the abdominal pain and ascites, abdominal paracentesis was performed. The 3 liters of ascites with haemorrhagic fluid were drained within a day.

Reviewer 2 Report

Comments and Suggestions for Authors

Thank you for your report

As a rare disease this case add knowledge on atypical presenbtation of this kind of renal neoplasm.

You hgave to add some relevant information:

- did the patient have any trauma?

-  any pro-haemorragic drug: medication? antiplatelet or other)

- it seems as a potential wunderlinch syndrome as acute presentation ( you can add some discussion on this topic)

- what was coagulative status of patient befroe and after surgery and particularly at time of the second laparotomy.

- do you have a dedicated uro-pathologis or alternatively usually ask/provide a second opinion in selected cases?

You should add data on discussion on pathological diagnosis possibility to avoid this misdiagnosis.

Author Response

Dear reviewer,
Thank you for your time and observations. We have made some changes in the main text according to your recommendations. 
This patient did not have any trauma and he did not use any drugs that could cause bleeding. 
We also added coagulative status before every procedure.
In our hospital, we do have a dedicated uropathologist team that evaluates the findings.
Thank you.